# The Role of the Second Extracellular Loop of Norepinephrine Transporter, Neurotrophin-3 and Tropomyosin Receptor Kinase C in T Cells: A Peripheral Biomarker in the Etiology of Schizophrenia

**DOI:** 10.3390/ijms22168499

**Published:** 2021-08-07

**Authors:** Daniela Rodrigues-Amorim, Marta Iglesias-Martínez-Almeida, Tania Rivera-Baltanás, Patricia Fernández-Palleiro, Luis Freiría-Martínez, Cynthia Rodríguez-Jamardo, María Comís-Tuche, María del Carmen Vallejo-Curto, María Álvarez-Ariza, Marta López-García, Elena de las Heras, Alejandro García-Caballero, Jose Manuel Olivares, Carlos Spuch

**Affiliations:** 1Translational Neuroscience Research Group, Galicia Sur Health Research Institute, CIBERSAM, Hospital Álvaro Cunqueiro, Bloque Técnico, Planta 2, Sala de Investigación, Estrada Clara Campoamor, 341, 36212 Vigo, Spain; daniela.amorim@sapo.pt (D.R.-A.); marta.iglesias@iisgaliciasur.es (M.I.-M.-A.); tania.rivera@iisgaliciasur.es (T.R.-B.); patricia.palleiro@iisgaliciasur.es (P.F.-P.); luis.freiria@iisgaliciasur.es (L.F.-M.); cynthia.rodriguez@iisgaliciasur.es (C.R.-J.); mariacomistuche@gmail.com (M.C.-T.); maria.del.carmen.vallejo.curto@sergas.es (M.d.C.V.-C.); maria.alvarez.ariza@sergas.es (M.Á.-A.); Marta.Lopez.Garcia@sergas.es (M.L.-G.); Elena.de.las.Heras.Linero@sergas.es (E.d.l.H.); alejandro.alberto.garcia.caballero@sergas.es (A.G.-C.); 2Translational Neuroscience Group, Universidade de Vigo, 36310 Vigo, Spain; 3Department of Psychiatry, Hospital Álvaro Cunqueiro, 36213 Vigo, Spain

**Keywords:** schizophrenia, neurotrophin-3, second extracellular loop of NET, tropomyosin receptor kinase C, T cells

## Abstract

The neurobiology of schizophrenia is multifactorial, comprising the dysregulation of several biochemical pathways and molecules. This research proposes a peripheral biomarker for schizophrenia that involves the second extracellular loop of norepinephrine transporter (NEText), the tropomyosin receptor kinase C (TrkC), and the neurotrophin-3 (NT-3) in T cells. The study of NEText, NT-3, and TrkC was performed in T cells and plasma extracted from peripheral blood of 54 patients with schizophrenia and 54 healthy controls. Levels of NT-3, TrkC, and NET were significantly lower in plasma and T cells of patients compared to healthy controls. Co-immunoprecipitation (co-IPs) showed protein interactions with Co-IP NEText–NT-3 and Co-IP NEText–TrkC. Computational modelling of protein–peptide docking by CABS-dock provided a medium–high accuracy model for NT-3–NEText (4.6935 Å) and TrkC–NEText (2.1365 Å). In summary, immunocomplexes reached statistical relevance in the T cells of the control group contrary to the results obtained with schizophrenia. The reduced expression of NT-3, TrkC, and NET, and the lack of molecular complexes in T cells of patients with schizophrenia may lead to a peripheral dysregulation of intracellular signaling pathways and an abnormal reuptake of norepinephrine (NE) by NET. This peripheral molecular biomarker underlying schizophrenia reinforces the role of neurotrophins, and noradrenergic and immune systems in the pathophysiology of schizophrenia.

## 1. Introduction

Schizophrenia is a chronic and multidimensional disorder, characterized by a particular psychopathology whose symptomatology comprises positive symptoms (hallucinations, delusion), negative symptoms (disorganized speech, social withdrawal), and impaired cognitive abilities [1,2]. The neurobiology of schizophrenia is multifaceted, involving the dysregulation of several biochemical pathways, as well as changes in the levels of various molecules, such as neurotrophins, neurotransmitters, or cytokines [3,4,5].

Neurotrophins are a family of polypeptide growth factors that play a crucial role in the central nervous system (CNS), specifically in neuronal development (neuronal survival, migration, and differentiation), synaptogenesis (synaptic plasticity and connections), and neuroplasticity (dendrite pruning and axon growth) [6,7,8,9]. In fact, the action of NTs depends on two classes of receptors, the tropomyosin receptors kinase (Trk) and the pan neurotrophin receptor (p75NTR) [10,11,12]. Moreover, NTs regulate neurotransmitter release and can enhance monoamine transporter and receptor activity [13]. Abnormal levels of neurotrophic factors are associated with neuronal maldevelopment and neuroplasticity aberrations [8,14].

In the same way, norepinephrine (NE) has been associated with the negative and cognitive symptoms of schizophrenia [15], being a modulator of brain function by interacting with the dopaminergic system [16]. Nerve terminals of the postganglionic sympathetic nervous system (SNS) are the primary source of NE, although peripheral blood nuclear cells (PBMC) also synthetize NE [17]. Accumulated evidence suggests NE could modulate the immune system, including T cells, as well as the noradrenergic signaling [18,19,20].

In turn, the immune system is also a protagonist in schizophrenia [15,16]. Currently, immunopathogenesis has emerged as a critical approach, suggesting that the dysregulation of the immune system is a potential etiopathological mechanism for schizophrenia [17,18]. T cells produce signaling molecules (e.g., cytokines) that are involved in neuroinflammatory processes promoting neurodegeneration and loss of grey matter, which lead to cognitive impairment in schizophrenia [18,19]. Moreover, T cells express functional receptors and transporters for neurotransmitters in their surface establishing a bidirectional communication between the immune system and the CNS [20,21].

In conclusion, T cells express receptors and specific proteins involved in signaling routes of the CNS associated with neuropsychiatric disorders [22]. On the other hand, several studies reported atypical levels of NTs in plasma or serum of patients with schizophrenia [23,24,25]. In this perspective, peripheral events are crucial for maintaining CNS functionality. This research proposes a peripheral mechanism that involves the second extracellular loop of NET (NEText), TrkC, and NT-3 in T cells and plasma, which may be dysregulated in schizophrenia. In short, this potential peripheral molecular biomarker reinforces the role of NT-3, TrkC, NET, and the immune system in the pathophysiology of schizophrenia.

## 2. Results

### 2.1. Demographic and Clinical Data Analysis

A case–control study was performed to compare a group of patients with schizophrenia and a healthy control group. Both groups were matched by age and gender, and there were no significant differences between groups regarding age (*p* = 0.1310) and gender (*p* = 1.0000) (Table 1). Neutrophil–lymphocyte ratio (NLR) is a predictor of systemic inflammation. Thus, the NLR was quantified in patients and controls, and there were no statistically significant differences between groups (*p* = 0.6881) (Table 1).

Furthermore, a correlation analysis was conducted to investigate the relationship between levels of NT-3, TrkC, or NET, and clinical variables, such as PANSS scores, age of illness onset, duration of illness, or NLR of the group of patients with schizophrenia. No statistically significant results were found (*p* > 0.05) (Appendix A).

### 2.2. Quantification of NET, TrkC, and NT-3 Levels

Levels of NT-3 were measured in plasma, whereas NET and TrkC levels were measured in T cells (Figure 1). The plasma levels of NT-3 were significantly lower in patients than in controls (111.7 ± 3.63 vs. 125.0 ± 2.54; *p* = 0.0040) after a nonlinear regression calculation (R^2^ = 0.9999). The analysis of NET (0.4809 ± 0.069 vs. 1.000 ± 0.10; *p* < 0.0001) and TrkC (0.6136 ± 0.072 vs. 1.000 ± 0.11; *p* = 0.0032) showed a substantial decrease in these proteins in T cells of patients compared to T cells of healthy controls.

### 2.3. Identification of an Interaction between the Second Extracellular Loop of NET and NT-3 or TrkC in T Lymphocytes

Co-immunoprecipitations (Co-IPs) were performed for analyzing protein–protein interactions between NEText, NT-3, and TrkC. Hence, immunocomplexes were analyzed by Western blot: Co-IP NEText-blot NT-3 (0.0672 ± 0.33 vs. 1.000 ± 026, *p* = 0.0003); and Co-IP NEText-blot TrkC (0.2286 ± 0.08 vs. 1.000 ± 0.23, *p* = 0.0020). The conformation of immunocomplexes in T cells was confirmed in the control group, but not in the group of patients with schizophrenia (Figure 2). Equally, the reverse Co-IPs were also performed and confirmed the interaction of NT-3–NEText (0.4257 ± 0.09 vs. 1.000 ± 0.08, *p* = 0.0021) and TrkC–NEText (0.3785 ± 0.10 vs.1.000 ± 0.25, *p* = 0.030) in the T cells of controls (Figure 2). To summarize, immunocomplexes reached statistical relevance in the control group (Figure 2). No statistical significance was found for NT-3 and TrkC Co-Ips and reverse Co-IPs in both groups (*p* > 0.05) (Figure 2).

### 2.4. Immunofluorescence Microscopy

Fluorescence microscopy images of T cells, double-labelled for NEText and NT-3 or NEText and TrkC, were studied. To support the idea of immunocomplex and protein interactions, immunofluorescence staining was performed for corroborating the presence of these proteins in T cells and in the extracellular space. TrkC and NEText appeared in the plasma membrane, whereas NT-3 had an extracellular distribution. T cells of patients with schizophrenia (*n* = 5) stained with NEText–NT-3 showed a restricted distribution of NEText and NT-3 compared with the controls (*n* = 5) (Figure 3a). In the same sense, the distribution of TrkC (NEText–TrkC staining) was restricted in T cells of patients with schizophrenia (Figure 3e). Intensity profile graphs revealed a distinctive distribution of proteins in patients with schizophrenia (red) and controls (green), confirming the results obtained by Co-IPs.

### 2.5. Characterization of NT-3–NEText and TrkC–NEText Docking Predictions

After finishing the job, the CABS-dock server provided a results webpage with the docking construct. Ten individual predictive models based on NT-3 protein (PBD code: 1nt3:A) or TrkC protein (PBD code: 1wwc:A), and CCCCCEECCCCCCCC peptide secondary structure were shown (Appendix A). The first model was the most important prediction considering the docking calculations. The cluster density (average difference between cluster elements and the number of elements) was 48.3648 and the average RMSD was 4.6935 Å for NT-3, whereas, for TrkC, the cluster density was 50.5506 and the average RMSD was 2.13647 Å (Figure 4).

## 3. Discussion

Over the last years, the dysregulation of the immune system has been linked to schizophrenia due to an abnormal pattern of proinflammatory cytokines and signs of peripheral inflammation detected in patients [26]. On the other hand, evidence suggests that immune cells, such as T cells, express monoamine and NT receptors in their surface [27,28,29]. NTs participate in fetal development processes (embryogenesis and organogenesis), nerve regeneration, and neuroplasticity in adulthood [30]. Specifically, NT-3 is crucial for neurite growth, differentiation, and neuronal survival, and also plays a role in early neurodevelopment, given that its levels increased in the fetal stage and decreased in adulthood [7,31]. Its receptor, TrkC, regulates different signaling pathways, such as extracellular signal-regulated kinases (ERKs) and phosphatidylinositol-3-kinase (PI3K)-protein kinase B (AKT) pathways that promote cell growth, differentiation, and survival, and phospholipase C-γ (PLCγ) signaling that is critical for synaptic plasticity (Figure 5) [32,33]. The neurotrophin hypothesis for schizophrenia claims that abnormal levels of these neurotrophic factors induce neurodevelopment and neuroplasticity disruptions [5,34].

Regarding NET, it has been considered a static protein for many years. Nevertheless, several studies have highlighted a dynamic regulation of NET by intracellular and extracellular molecules [18,35,36]. NET expression is regulated by growth factors, such as NT-3, transforming growth factor β1 (TGF-β1), and fibroblast growth factor-2 (FGF-2) [37,38]. Norepinephrine (NE) is transported into neurons by NET, a Na^+^/Cl^−^ dependent neurotransmitter transporter whose blockade increases extracellular levels of NE (Figure 5) [39,40,41]. Throughout the years, the noradrenergic hypothesis has acquired consistent evidence in the pathogenesis of schizophrenia [42,43].

Considering that NT-3 regulates the expression of monoamines, such as NE, and T cells express both NT receptors (e.g., TrkC) and monoamine transporters (e.g., NET), it is essential to understand what happens in a psychopathological disorder such as schizophrenia. Therefore, in this study, the levels of NET, TrkC, and NT-3 were measured in T cells and plasma, respectively. The levels of the three molecules were significantly reduced in patients with schizophrenia compared to controls. Analyzing Co-IPs, it was found that NT-3 immunoprecipitated with NEText in T cells of controls, as well as TrkC, which indicates the presence of molecular complexes in immune cells of controls in opposition to the results obtained in the T cells of patients with schizophrenia. Moreover, immunofluorescence staining of T cells was performed to analyze the distribution of NET, TrkC, and NT-3, the results of which corroborated the idea of a disrupted molecular finding underlying schizophrenia (Figure 5). Clinically, levels of NET, NT-3, and TrkC were correlated with PANSS scores, illness onset, duration of illness, and NLR of patients with schizophrenia. However, the expression of proteins was not significantly associated with any of the clinical variables (*p* > 0.05) (Appendix A).

The computational modeling of protein–peptide docking allowed the study of the interactions between NT-3 or TrkC and the peptide LLNGSVLGNHTKYSK, corresponding to amino acid residues 189–204 of human NET (anti-NEText immunogen). The results obtained showed a high accuracy prediction for the TrkC–NEText complex and a medium accuracy prediction for the NT-3–NEText complex. The molecular docking study confirmed the target specificity between proteins and peptide, supporting the previous results on the formation of NT-3 controlling neurogenesis through the activation of TrkC that regulates many signaling pathways, NE synthesis, and synaptic activity in the CNS [8,44]. In accordance with our results, this cascade of events also occurs in the periphery, more precisely, in T cells. The establishment of molecular complexes between NT-3 or TrkC and the second extracellular loop of NET in T cells of healthy controls explains a peripheral mechanism where NET is an activity-dependent transporter. NT-3 activates TrkC receptors and, after the complex internalization, different signaling pathways, such as PI3K/AKT, extracellular signal-regulated kinase (ERKs), Ras/mitogen-activated protein kinase (MAPK), and phospholipase C-γ (PLC-γ), initiate a cascade of intracellular signal transduction [45,46]. In turn, NET exerts a dynamic control of extracellular NE concentration through reuptake of the NE released in the synaptic cleft, playing an important role in pre/postsynaptic homeostasis [18,47]. Evidence suggests that NT-3 regulates NET expression, for example, in cultured quail neural crest cells [38]. Thus, NT-3 modulates the synaptic transmission of NE and increases NE reuptake [48]. Based on this evidence, the regulatory mechanism that involves the second extracellular loop of NET, TrkC, and NT-3 depends on the molecular complexes formed, which are essential for the appropriate activation of the signal transduction. This potential peripheral biomarker reinforces the role of NT-3, TrkC, NET, and the immune system in the pathophysiology of schizophrenia.

## 4. Materials and Methods

### 4.1. Subjects and Samples

Venous blood samples from 54 patients diagnosed with schizophrenia and 54 healthy controls were collected in vacuum tubes containing K2EDTA between 7:00 and 9:00 h. The samples of these patients and control subjects were obtained from the Álvaro Cunqueiro Hospital in Vigo, Spain. Psychiatrists, based on the fifth edition of the *Diagnostic and Statistical Manual of Mental Disorders* (DSM-5), established the diagnosis of schizophrenia, for which the antipsychotics treatment is described in Appendix A. The study’s inclusion criteria were patients with schizophrenia over 18 years old who provided a signed informed consent compliant with the guidelines of the Helsinki Declaration and approved by the Ethics Committee (Galician Network of Research Ethics Committees, Registration code: 2019/120). Subjects with other psychiatric disorders, neurological disorders or traumatic brain injury, infectious disease or immune/autoimmune conditions, or a history of substance abuse were excluded from the study. Pregnant or breastfeeding women were also excluded. Patient cohorts were matched by age and sex with healthy volunteers. The control group was also requested to provide a signed informed consent.

### 4.2. Clinical and Demographic Assessment

A demographic questionnaire was administered to all patients upon their admission to the Psychiatry Unit of the Álvaro Cunqueiro Hospital, and to healthy volunteers. The positive and negative syndrome scale (PANSS) was also used to evaluate the intensity of their positive and negative symptoms, as well as the general psychopathology of the patients with schizophrenia. Both instruments were applied by a trained psychologist independent from the study. Lymphocyte and neutrophil counts were recollected from clinical records. Clinical and demographic characteristics are showed in Table 1.

### 4.3. Sample Preparation

Collected peripheral blood was diluted 1:1 in phosphate buffer saline (PBS, homemade). Peripheral blood mononuclear cell (PBMC) isolation was performed by density gradient centrifugation with Ficoll-Paque™ (Amersham Biosciences, Freiburg, Germany) with 6% of dextran (Sigma-Aldrich, St. Louis, MO, USA) at 2000 rpm for 35 min at room temperature (RT) to a complete stop without braking. Cells were transferred to a Falcon tube with filtered PBS and centrifuged at 1500 rpm for 5 min at RT. Supernatant was discarded and the pellet was resuspended in 3 mL of RPMI 1640 medium (Gibco, Thermo Fisher Scientific, Waltham, MA, USA) supplemented with 10% of fetal bovine serum (FBS) (Gibco, Thermo Fisher Scientific, Waltham, MA, USA) and 1% of penicillin/streptomycin (Gibco, Thermo Fisher Scientific, Waltham, MA, USA). Samples were stored using CoolCell^®^ containers and, after 24 h, they were relocated in their respective box at −80 °C.

### 4.4. Immunoprecipitation (IP) and Western Blot

Cell samples (100 µL) were prepared using a lysis buffer (Tris-HCL 50 mM; NaCL 120 mM; IGEPAL 0.5%; NaF 100 mM; NaVO3 2mM) and incubated with the primary antibody overnight at 4 °C in a rotating mixer: anti-NET extracellular rabbit polyclonal antibody 1:500 (AMT-002, Alomone Labs, Jerusalem, Israel), TrkC rabbit polyclonal antibody 1:500 (ab227289, Abcam, Cambridge, UK), or anti-NT3 rabbit polyclonal antibody 1:500 (ANT-003, Alomone Labs, Jerusalem, Israel). Then, 100 µL of protein A (Protein A/G Plus Agarose, Thermo Fisher Scientific, Inc., Waltham, MA, USA) was added to samples, which were incubated for 2 h at 4 °C in a rotating mixer. After, samples were centrifuged at 14,000 rpm for 1 min at 4 °C, the supernatant was carefully aspirated, and 100 µL of lysis buffer was added. Samples were again centrifuged and the process was repeated 2 times. Finally, samples were mixed with 2× volume of Laemmli buffer 2× (Bio-Rad, Irvine, CA, USA) (950 µL of Laemmli buffer and 50 µL of β-mercaptoethanol) and boiled at 95 °C for 5 min. Fraction samples (20 µL) were loaded in 6–10% Bis-Tris polyacrylamide gels, and an electrophoresis was performed in a PowerPac™ universal power supply (Bio-Rad, CA, USA) at 60 V for 30 min, and at 100 V for about 120 min. The proteins were immediately transferred to polyvinylidene difluoride membranes (Immun-Blot^®^ PVDF membrane, Bio-Rad, CA, USA) contained in a PowerPac™ universal power supply (Bio-Rad, CA, USA) set at 0.25 A for 60 min (for two gels). The membranes were blocked with 5% milk or 5% bovine serum albumin (BSA) in a Tris-buffered saline solution with Tween (TBST) for 20 min, and washed three times with the same TBST solution. The membranes were then incubated overnight at 4 °C over stirrers with the primary antibody: anti-NET extracellular rabbit polyclonal antibody 1:1000 (AMT-002, Alomone Labs, Jerusalem, Israel), TrkC rabbit polyclonal antibody 1:1000 (sab1306628, Sigma-Aldrich, St. Louis, MO, USA), anti-NT3 rabbit polyclonal antibody 1:1000 (WB) (ANT-003, Alomone Labs, Jerusalem, Israel), anti-phosphotyrosine (p-Tyr) mouse monoclonal antibody 1:1000 (ab10321, Abcam, Cambridge, UK), or anti-β-actin rabbit polyclonal antibody 1:14,000 (600-401-886, Rockland Immunochemicals, Limerick, PA, USA). After washing them three times with TBST, the membranes were incubated with the appropriate secondary rabbit or mouse antibody 1:10,000 (GE Healthcare Life Sciences, UK) for 60 min over a stirrer. The membranes were then washed again twice with TBST and once with TBS. The ChemiDoc XRS+ system (Bio-Rad, CA, USA) was then used to analyze the chemiluminescence of the membranes using Immobilon Forte Western HRP substrate (Merck, Kenilworth, NJ, USA). Image Lab 6.0 software (Bio-Rad, Irvine, CA, USA) was used to analyze the blot images acquired, and a densitometric band quantification was performed by means of ImageJ (Fiji). The housekeeping protein β-actin was used as a loading control.

### 4.5. Immunofluorescence Microscopy

Fresh T cells were treated with EnVision FLEX (Dako, Denmark) for antibody visualization according to the manufacturer’s instructions. We performed five analyses: (1) 1:100 anti-CD3 mouse monoclonal antibody FITC conjugated (MA1-10178, Thermo Fisher Scientific, Inc., Waltham, MA, USA) and 1:250 anti-NET extracellular rabbit polyclonal antibody (AMT-002, Alomone Labs, Jerusalem, Israel); (2) 1:100 anti-CD3 mouse monoclonal antibody FITC conjugated (MA1-10178, Thermo Fisher Scientific, Inc., Waltham, MA, USA) and 1:500 anti-NT3 rabbit polyclonal antibody (ANT-003, Alomone Labs, Jerusalem, Israel); (3) 1:100 anti-CD3 mouse monoclonal antibody FITC conjugated (MA1-10178, Thermo Fisher Scientific, Inc., Waltham, MA, USA) and 1:250 anti-TrkC rabbit polyclonal antibody (sab1306628, Sigma-Aldrich, St. Louis, MO, USA); (4) 1:250 anti-NET extracellular rabbit polyclonal antibody (AMT-002, Alomone Labs, Jerusalem, Israel) and 1:100 anti-NT3 mouse monoclonal antibody (TA500030, Thermo Fisher Scientific, Inc., Waltham, MA, USA); and (5) 1:250 anti-NET extracellular rabbit polyclonal antibody (AMT-002, Alomone Labs, Jerusalem, Israel) and 1:100 anti-TrkC (CP10188, Cell Applications, Inc., San Diego, CA, USA). Samples were incubated with primary antibody overnight at 4 °C in a rotating mixer. Then, samples were washed with 1% Tris-buffered saline (TBS) and incubated for 2 h at 4 °C in a rotating mixer with 1:250 fluorescence-conjugated secondary antibodies (A-21202 and R27117 Alexa Fluor™, Thermo Fisher Scientific, Inc., Waltham, MA, USA). Finally, they were mounted in Superfrost^®^ Plus Slides (Thermo Fisher Scientific, Inc., Waltham, MA, USA) with Vectashield (H-1000, Vector Laboratories, Ldt., Peterborough, UK). Micrographs were taken by a fluorescent microscope Leica DMI 6000 HCX PL APO CS 63.0 × 1.40 OIL UV objective that was controlled by Leica LAS X software. Quantification was performed by means of ImageJ 1.50i (NIH, Bethesda, USA) (Appendix A).

### 4.6. NT-3 Plasma Quantification

NT-3 concentration of plasma samples of patients and healthy controls was quantified by sandwich enzyme-linked immunosorbent assay (ELISA), using a commercial kit (Human Neurotrophin 3 ELISA Kit, ab100615, Lote No. GR3229491-1, Abcam, Cambridge, UK), according to the manufacturer’s instructions. Tests were performed in duplicate, and an automated microplate reader (Biochrom ASYS UVM 340, Cambridge, UK) measured the optical density at 450 nm with Mikrowin 2000 software (Berthold Technologies, Germany). Plasma total proteins were measured by a bicinchoninic acid assay (BCA, Pierce Chemicals, Rockford, IL, USA).

### 4.7. NT-3 or TrkC Proteins and NEText Peptide Docking Prediction

CABS-dock web server was used to determine the structure of the protein–peptide complex (http://biocomp.chem.uw.edu.pl/CABSdock/, accessed on 15 September 2020). Computational modeling of protein–peptide docking was performed using CABS-dock server, uploading the following inputs: protein 1nt3 Protein Data Bank (PBD) code or 1wwc PDB code, both obtained in PBD in Europe (https://www.ebi.ac.uk/pdbe/, accessed on 1 September 2020); and the peptide sequence LLNGSVLGNHTKYSK (part of the second extracellular loop of human NET corresponding to amino acid residues 189–204, detected by the antibody anti-NEText), whose secondary structure was assigned by PSIPRED v4.0 (https://mobyle.rpbs.univ-paris-diderot.fr/cgi-bin/portal.py#forms::psipred, accessed on 15 September 2020) as CCCCCEECCCCCCCC through the job K13689290838957. The identification parameters were introduced: protein chain identifier (A), peptide identifier (PEP), cut-off (5.0 Angstroms). The tridimensional structure of protein and the peptide conformation were constructed by the workflows represented in Appendix A. The computational modeling of protein–peptide interactions was based on the prediction of the protein binding site and the docking of the peptide to the binding site [17]. Protein structure was acquired by X-ray diffraction with a resolution of 2.40 Å for NT-3 and 1.90 Å for TrkC. The results provided 10 binding models, and the model with the highest number of cluster elements and lower ligand root-mean-square deviation (L-RMSD) (high accuracy: L-RMSD < 3 Å; medium accuracy: 3 Å ≤ L-RMSD ≤ 5.5 Å; and low accuracy: L-RMSD > 5.5 Å) was selected (Appendix A) [17].

### 4.8. Statistical Analysis

The GraphPad Prism 7 software (GraphPad Software Inc., San Diego, CA, USA) was used to manage the resulting data and to perform the statistical analysis. The mean age of both cohorts was compared with the Mann–Whitney U-test, and the differences between sex ratios were analyzed with Fisher’s exact test. A two-sample Student’s *t*-test was used to analyze differences between the schizophrenia group and control group. In addition, the correlation between the PANSS scores, duration of illness, age of illness onset, and neutrophil–lymphocyte ratio (NLR) were analyzed by means of a Pearson correlation analysis. A nonlinear regression calculation was used to fit the curve of NT-3 that was measured by ELISA using the Curve Expert software. The detection of the housekeeping protein β-Actin was used for the normalization of the levels of NEText and TrkC, whereas total protein measured by BCA was applied for the normalization of NT-3 levels. Statistically significant results are assumed considering a *p*-value ≤ 0.05.

## 5. Conclusions

Following the results, substantial alterations in the expression of the NT-3, TrkC, and NET in T cells of patients with schizophrenia were found. Based on the imperative regulation of neurotrophins and their receptors on the noradrenergic system, we propose a potential peripheral molecular biomarker underlying schizophrenia, whose reduced expression of NT-3, TrkC, and NET, and the lack of molecular complexes in T cells may lead to a peripheral dysregulation of the signaling pathways and an abnormal reuptake of NE by NET. The findings suggest that a molecular regulatory mechanism described in neurons is also present in T cells, justifying a potential peripheral biomarker of schizophrenia based on the immune, neurotrophic, and noradrenergic systems.

## Figures and Tables

**Figure 1 ijms-22-08499-f001:**
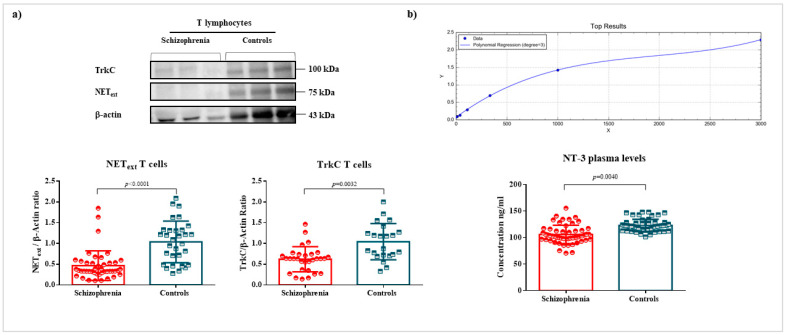
Levels of NET and TrkC in T cells, and NT-3 in plasma. Legend: quantification of NET and TrkC levels in T cells and NT-3 in plasma, and their respective scatter plots and bar graphs (mean ± SD). (**a**) NEText (45 patients with schizophrenia vs. 31 controls) and TrkC (31 patients with schizophrenia vs. 22 controls) levels were measured by Western blot and normalized with the housekeeping protein β-actin. Relative units refer to the mean of control values after normalization. The results showed a reduction in NET levels (*p* < 0.0001) and TrkC levels (*p* = 0.0032) in T cells of patients with schizophrenia compared to the control group. (**b**) The analysis of NT-3 (54 patients with schizophrenia and 54 healthy controls) in plasma revealed a reduction in this neurotrophin in the plasma of patients in comparison with controls (*p* = 0.0040), whose concentration was measured by ELISA using a polynomial regression method (Y = absorbance; X = concentration, degree = 3; R^2^ = 0.9999).

**Figure 2 ijms-22-08499-f002:**
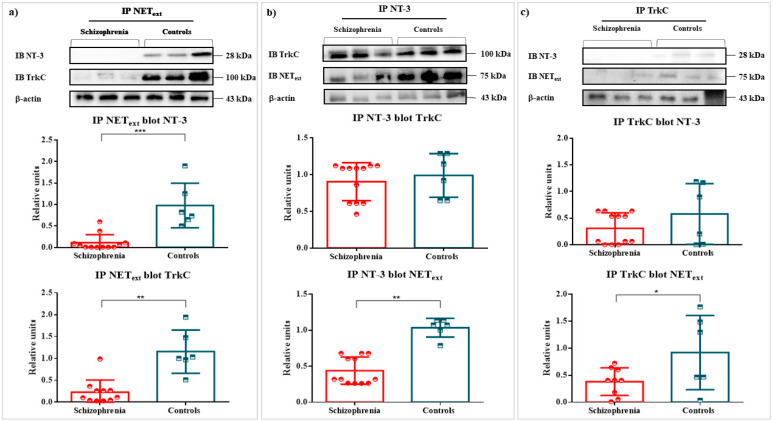
Co-IPs of NEText and NT-3 or TrkC in T cells of patients with schizophrenia and healthy controls. Legend: Co-IPs of NEText, NT-3, and TrkC in T lymphocytes of patients with schizophrenia and controls. Legend: Measurement of NEText and TrkC levels in T cells and NT-3 levels in plasma, and their respective scatter plots and bar graphs (mean ± SD). (**a**) Co-IP NEText-blot NT-3 (12 patients with schizophrenia vs. 6 controls; *p* = 0.0003) and Co-IP NEText-blot TrkC (10 patients with schizophrenia vs. 6 controls *p* = 0.0020). (**b**) Co-IP NT-3-blot TrkC (12 patients with schizophrenia vs. 6 controls; *p* > 0.05) and Co-IP NT-3-blot NEText (12 patients with schizophrenia vs. 6 controls; *p* = 0.0021). (**c**) Co-IP TrkC-blot NT-3 (12 patients with schizophrenia vs. 6 controls; *p* > 0.05) and Co-IP TrkC-blot NEText (9 patients with schizophrenia vs. 6 controls; *p* = 0.030) (* *p* < 0.05; ** *p* < 0.01 and *** *p* < 0.001).

**Figure 3 ijms-22-08499-f003:**
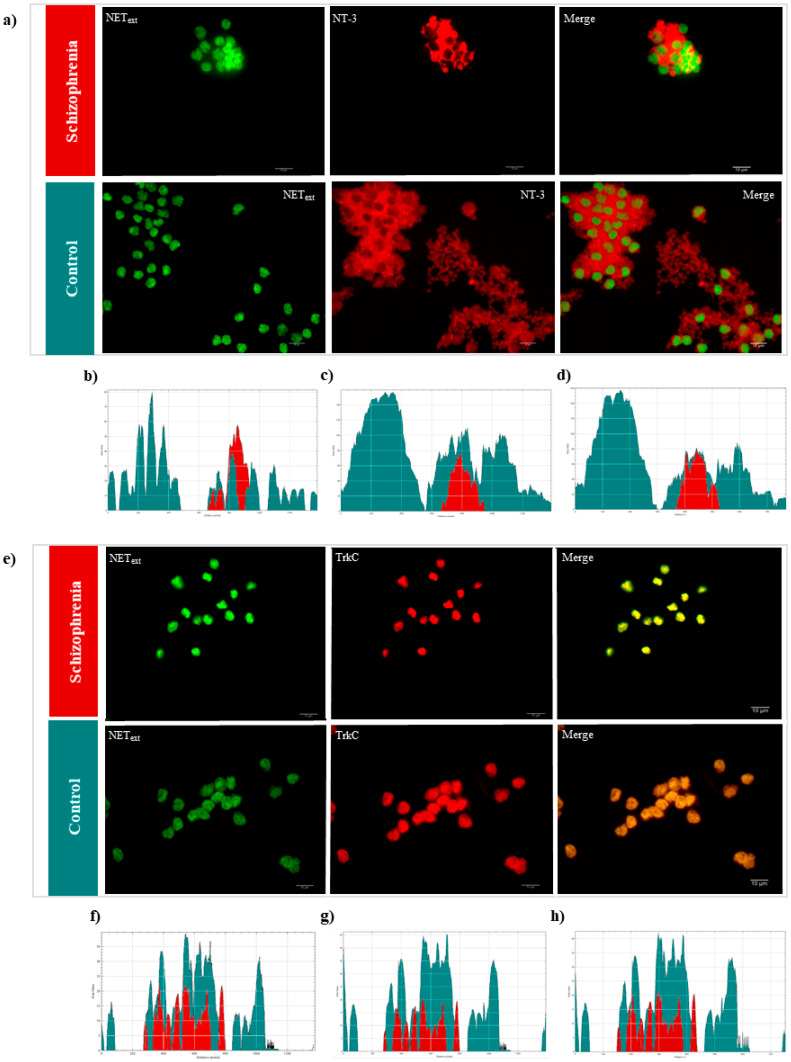
Fluorescence microscopy images of blood T lymphocytes. Legend: fluorescence microscopy images of blood T lymphocytes. T cells were stained with anti-NEText + anti-Alexa Fluor 488 (green color) and anti-NT-3 + anti-Alexa Fluor 594 (red color) antibodies; or anti-NEText + anti-Alexa Fluor 488 (green color) and anti-TrkC + anti-Alexa Fluor 594 (red color) and expanded from peripheral blood mononuclear cells with an HCX PL APO CS 63.0 × 1.40 OIL UV objective. The white bar represents 10 µm. (**a**) NEText–NT-3 staining of T lymphocytes of patients with schizophrenia and controls. (**b**–**d**) Intensity profile graphs (distance in pixels vs. intensity in grey-scale values) of T cells of patients with schizophrenia (red) and controls (green): (**c**) distribution of NEText, (**d**) distribution of NT-3, and (**e**) distribution of merged results. NEText–TrkC staining of T lymphocytes of patients with schizophrenia and controls. (**f**–**h**) Intensity profile graphs (distance in pixels vs. intensity in grey-scale values) of T cells of patients with schizophrenia (red) and controls (green): (**f**) distribution of NEText, (**g**) distribution of TrkC, and (**h**) distribution of merged results.

**Figure 4 ijms-22-08499-f004:**
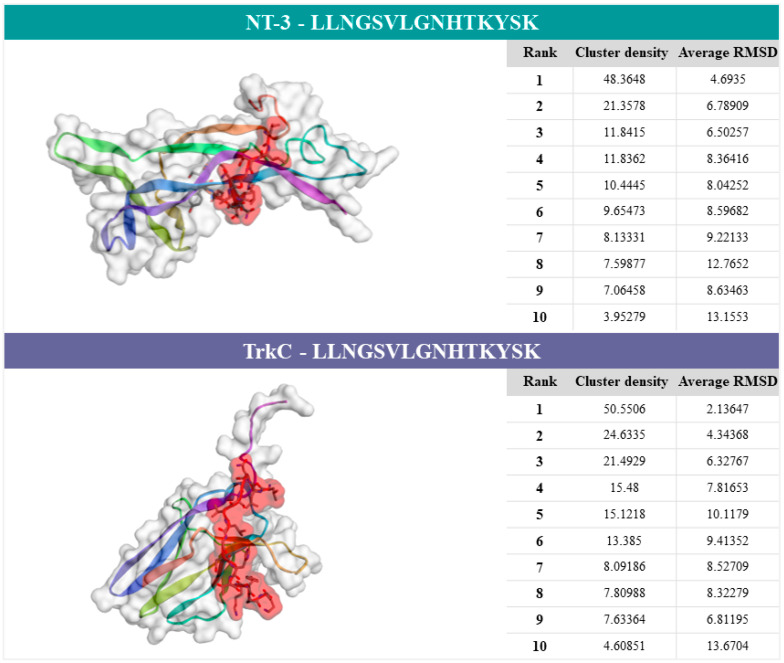
Predictive results of the CABS-dock server. Legend: computational modelling of protein–peptide docking of NT-3 (1nt3:A) or TrkC (1wwc:A) and the peptide sequence (part of the second extracellular loop of NET: LLNGSVLGNHTKYSK). The docking prediction figures of proteins–peptide correspond to model 1. Clustering details of the ten models are shown in the respective tables.

**Figure 5 ijms-22-08499-f005:**
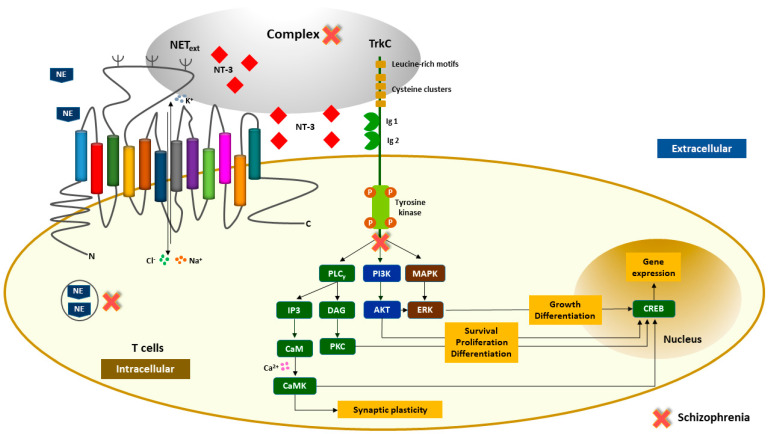
Schematic representation of NEText, NT-3, and TrkC distribution in T cells of patients with schizophrenia and healthy controls. Legend: NT-3-mediated signal transduction pathways through dimerization of TrkC receptors, which results in an intracellular activation of extracellular signal-regulated kinases (ERKs), phosphatidylinositol-3-kinase (PI3K)-protein kinase B (AKT), and phospholipase C-γ (PLCγ) pathways promoting cell survival, proliferation, differentiation, synaptic plasticity, and gene expression in T cells. The molecular complex between the second extracellular loop of NEText (corresponding to amino acid residues 189–204) and NT-3 or TrkC seems to be essential for appropriate activation of the signaling cascade in a peripheric context that involves immune system cells. The red cross represents the mechanisms/pathways that are disrupted in schizophrenia.

**Table 1 ijms-22-08499-t001:** Clinical and demographic characteristics of participants. Legend: ^1^ Mann–Whitney U-test *p*-value; ^2^ Fisher’s exact test *p*-value; ^3^ unpaired two-tailed *t*-test; ^4^ years. PANSS: positive and negative syndrome scale; PANSS positive and negative ranges: 7–49; PANSS general range: 16–112; PANSS total range: 28–210.

Parameter	Schizophrenia	Controls	*p*-Value
**Total Number** (*N*)	54	54	-
**Age** (mean ± SD)	39.93 ± 12.29	44.61 ± 12.84	0.1310 ^1^
**Gender** (M/F)	30/24	31/23	1.0000 ^2^
**Neutrophil–lymphocyte ratio**	2.57 ± 1.29	2.81 ± 1.84	0.6881 ^3^
**Illness onset** (mean ± SD) ^4^	29.69 ± 9.70	-	-
**Duration of illness** (mean ± SD) ^4^	10.22 ± 8.14	-	-
**PANSS** (mean ± SD)			
**PANSS Positive**	20.85 ± 5.44	-	-
**PANSS Negative**	38.59 ± 6.67	-	-
**PANSS General**	36.81 ± 7.36	-	-
**PANSS Total**	85.89 ± 13.95	-	-

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
