# Peer review of "The Role of the Second Extracellular Loop of Norepinephrine Transporter, Neurotrophin-3 and Tropomyosin Receptor Kinase C in T Cells: A Peripheral Biomarker in the Etiology of Schizophrenia"

_ijms, 2021, doi:10.3390/ijms22168499_

Round 1

Reviewer 1 Report

Journal Title: International Journal of Medical Sciences

Article Type: Original Research

Article Title: The role of the second extracellular loop of NET, NT-3 and TrkC receptor in T cells: a peripheral pathomolecular mechanism of schizophrenia

Dear Editor,

Dear Authors,

This manuscript gives a new insight into noradrenergic-neurotrophic-immune interactions in schizophrenia. This clinical sample is representative, the methodology is adequate, and the analysis is presented in detail. It is well written, comprehensive, and presented with illustrative tables, graphs, and figure.

I would propose only minor changes:

- Title: Avoid using the abbreviations in the Title or maybe to be more prominent in pointing out the connection of T cells and an abnormal reuptake of NE. Be more precise what is extracellular loop and what is in the T cell in the whole manuscript, in some part it is pretty confusing (“Levels of NT-3 were measured in plasma, whereas NEText and TrkC levels were measured in T cells.”).

- Abstract: Explain all abbreviations.

- Introduction: Consider extending this section with data describing especially noradrenaline: the connection of noradrenaline and T cells, noradrenaline and NT-3 and specific role of noradrenaline in schizophrenia.

- Subjects and samples: Clearly state the phase of the disease and applied therapy, which all could impact on the parameters that were measured.

- Limitations of the study: Some procedures were not performed on the whole sample, explain this and other possible limitations.

- It is usual to statistical significance be noted with small letter and Italic (p).

Author Response

Reviewer #1

We are grateful that referee appreciated the significance of our work and are thankful for his/her comments. The revised version of our manuscript presents novel elucidations on the issues raised by the reviewers.

All changes made to the text are highlighted (yellow). Please find below the point-to point clarifications.

“This manuscript gives a new insight into noradrenergic-neurotrophic-immune interactions in schizophrenia. This clinical sample is representative, the methodology is adequate, and the analysis is presented in detail. It is well written, comprehensive, and presented with illustrative tables, graphs, and figure.”

We thank this referee for his/her comment.

Minor changes:

- “Title: Avoid using the abbreviations in the Title or maybe to be more prominent in pointing out the connection of T cells and an abnormal reuptake of NE. Be more precise what is extracellular loop and what is in the T cell in the whole manuscript, in some part it is pretty confusing (“Levels of NT-3 were measured in plasma, whereas NEText and TrkC levels were measured in T cells.”).”

            We thank this comment. Abbreviations were placed due to the length of the title, but we agree they should be avoided. Title was rewritten without abbreviations. Regarding what is measured in plasma or T cells, receptors and transporters are located in the plasma membrane, so their expression comes from cell membranes, in this case T cell. Neurotrophin 3 is an extracellular protein (extracellular region or secreted), and that is why we measure it in plasma. Although the location of the molecules is something patented in the literature, in the manuscript, it is also described in the section 2.5: Line 148: “TrkC and NEText appeared in the plasma membrane, whereas NT-3 had an extracellular distribution.” We believe that in this sense, the confusion can be dissipated.

- “Abstract: Explain all abbreviations”.

We thank this comment. The respective abbreviation names have been added: Co-immunoprecipitation (co-IPs) and norepinephrine (NE).

- “Introduction: Consider extending this section with data describing especially noradrenaline: the connection of noradrenaline and T cells, noradrenaline and NT-3 and specific role of noradrenaline in schizophrenia.”

We thank this suggestion. The following information has been added to the introduction: Line 53: “In parallel, norepinephrine (NE) has been related with negative and cognitive symptoms in schizophrenia [1]. As a neuromodulator of brain function that interacts with the dopaminergic system, its role in the pathophysiology of schizophrenia as growing over the years [2]. Nerve terminals of the post-ganglionic sympathetic nervous system (SNS) are the primary source of NE, although peripheral blood nuclear cells (PBMC) also synthetize NE [3]. Accumulated evidence suggests NE could modulate the immune system, including T cells, as well the noradrenergic signalling [4–6].”

- “Subjects and samples: Clearly state the phase of the disease and applied therapy, which all could impact on the parameters that were measured.”

We thank this comment. In Supplementary Material there is a description of the stages, number of patients, percentages and antipsychotic treatment (Supplementary Table 1). Näive corresponds to first-episode psychosis (FEP) and patients treated with antipsychotics were in a chronic phase. Moreover, the relationship between levels of NT-3, TrkC or NEText, and clinical variables (age of illness onset, duration of illness, Positive and Negative Syndrome Scale scores, and neutrophil-lymphocyte ratio, which is a predictor of systemic inflammation) was analysed, and no statistically significant results were found (P>0.05) (Supplementary Table 2).

- “Limitations of the study: Some procedures were not performed on the whole sample, explain this and other possible limitations.”

We thank this comment. Co-IPs and IF microscopy were used to examine protein complexes in patients with schizophrenia vs controls. Both are efficient methods to understand protein interactions, but they were unperformed on the whole sample. In this manuscript, we try justifying a peripheral pathomolecular mechanisms underlying schizophrenia based on the immune, neurotrophic and noradrenergic systems. Different techniques/approaches (Co-IPs, IF, CABS-dock) were used to prove steps-by-step, the statement described. This is not a methodological limitation but a study design (experimental procedure), whose exhaustive methodology described allow experimental replication (human cell, cell lines or animal models).

Examples of studies:

  1. Alfieri, A.; Sorokina, O.; Adrait, A.; Angelini, C.; Russo, I.; Morellato, A.; Matteoli, M.; Menna, E.; Erba, E.B.; McLean, C.; et al. Synaptic interactome mining reveals p140cap as a new hub for PSD proteins involved in psychiatric and neurological disorders. Front. Mol. Neurosci. 2017, 10, 1–15, doi:10.3389/fnmol.2017.00212.
  2. Fryland, T.; Christensen, J.H.; Pallesen, J.; Mattheisen, M.; Palmfeldt, J.; Bak, M.; Grove, J.; Demontis, D.; Blechingberg, J.; Ooi, H.S.; et al. Identification of the BRD1 interaction network and its impact on mental disorder risk. Genome Med. 2016, 8, 1–20, doi:10.1186/s13073-016-0308-X.

- “It is usual to statistical significance be noted with small letter and Italic (p).”

We thank this suggestion. p-value was rewritten with small letter and Italic (in both text and Figure 1).

Reviewer 2 Report

Major comment -

  1. It is hard to buy and thus, authors need to tone down that it is a peripheral pathomolecular mechanism of schizophrenia, rather it can just be a peripheral pathology of schizophrenia due to major organs affected or consequence of other molecular mechanism of schizophrenia.
  2. I completely missed the roles of NET, NT-3 and TrkC 2 receptor in T cells, rather in any immune cell and their effects on immune functions. Authors needs to potentially highlight, speculate and discuss it based on available literature.
  3. The specificity of antibody staining must be shown in T cells.

Minor comment -

  1. Number of samples/donors is missing in all figures. Is it 54 for controls and patients in all figures? Must be added to individual figures and sub-figures in the figure legends.
  2. What is the normality distribution of the data? Is student's t-test used for all comparisons?
  3. The box and whiskers plot must be combined with all data points included for clarity. See example attached. Were outliers removed before statistical analysis?
  4. Please define the box and whiskers plot in at lease first figure legend.

--

Best regards

Author Response

Reviewer 2

We are grateful that referee appreciated the significance of our work and are thankful for his/her comments. The revised version of our manuscript presents novel elucidations on the issues raised by the reviewers.

All changes made to the text are highlighted (yellow). Please find below the point-to point clarifications.

Major comment -

  1. “It is hard to buy and thus, authors need to tone down that it is a peripheral pathomolecular mechanism of schizophrenia, rather it can just be a peripheral pathology of schizophrenia due to major organs affected or consequence of other molecular mechanism of schizophrenia.”

We thank this referee for his/her comment. We do not sell anything. Based on diverse laboratorial techniques and protein-peptide interactions analysis, supporting data and additional data (supplementary material), we justify a peripheral pathomolecular mechanism underlying schizophrenia. Regarding the statement “just be a peripheral pathology of schizophrenia due to major organs affected”, if it means schizophrenia may have a peripheral etiology, whose major organ affected is the brain, we cannot exclude this hypothesis as it is a multifactorial disorder with an etiology is poorly understood (discussed in the manuscript: immunopathogenesis has emerged as a critical approach suggesting that the dysregulation of the immune system is a potential etiopathological mechanism for schizophrenia”). In another sense, we do not understand this point of view or which “major organs” are affected in schizophrenia. On the other hand, there are other molecular mechanisms in schizophrenia described in the literature, however, the conformation of complexes in T cells, the peripheral dysregulation of intracellular signalling pathways and an abnormal reuptake of NE by NET have never been described.

  1. I completely missed the roles of NET, NT-3 and TrkC 2 receptor in T cells, rather in any immune cell and their effects on immune functions. Authors needs to potentially highlight, speculate and discuss it based on available literature.

We thank this referee for his/her comment. Line 236: The establishment of molecular complexes between NT-3 or TrkC and the second extracellular loop of NET in T cells of healthy controls explains a peripheral mechanism where NET is an activity-dependent transporter. NT-3 actives TrkC receptors and after the complex internalization, different signalling pathways, such as PI3K/AKT, extracellular signal-regulated kinase (ERKs), Ras/mitogen activated protein kinase (MAPK) and phospholipase C-γ (PLC-γ) initiate a cascade of intracellular signal transduction [47,48]. In its turn, NET exerts a dynamic control of extracellular NE concentration through reuptake the NE released in the synaptic cleft, playing an important role in pre/post-synaptic homeostasis [36,49]. Evidence suggests that NT-3 regulates NET expression, for example, in cultured quail neural crest cells [40]. Thus, NT-3 modulates the synaptic transmission of NE and increase NE reuptake [50]. Based on these evidences, the regulatory mechanism that involves the second extracellular loop of NET, TrkC and NT-3 depends on the molecular complexes formed, which are essential for the appropriate activation of the signal transduction. This peripheral pathomolecular mechanism reinforces the role of NT-3, TrkC, NEText and the immune system in the pathophysiology of schizophrenia.”

Whit more detail, the cascade of events in the T cell can be seen in Figure 5.

  1. “The specificity of antibody staining must be shown in T cells.”

We thank this referee for his/her comment. We think it refers to the specificity of anti-CD3. Thus, anti-CD3 antibody is the T cell specific marker “1:100 anti-CD3 mouse monoclonal antibody FITC conjugated (MA1-10178, Thermo Fisher Scientific, Inc., Waltham, MA, USA)”. The immunostaining of T cells (green) is in the Supplementary Figure 3.

Minor comment -

  1. “Number of samples/donors is missing in all figures. Is it 54 for controls and patients in all figures? Must be added to individual figures and sub-figures in the figure legends.”

We thank this suggestion. The number of samples used for each experiment/figure was added to figure legends. In the immunofluorescence microscopy section, there is described in the text: T cells of patients with schizophrenia (n=5) stained with NEText-NT-3 showed a lower distribution of NEText and NT-3 comparing with controls (n=5)”. In Supplementary Material file, all original blots are identified with the samples used (schizophrenia or control).

  1. “What is the normality distribution of the data? Is student's t-test used for all comparisons?”

Section 4.8. (methodology):

  • Normalization of NEText and TrkC levels: housekeeping protein β-Actin in relative units
  • Normalization of plasma NT-3 levels: total protein measured by BCA in relative units
  • Age of both cohorts: Mann-Whitney U test
  • Sex ratios: Fisher’s exact test
  • Differences between schizophrenia and control groups: two-sample Student’s t-test
  • Correlation of proteins levels and clinical variables: Pearson correlation
  • Elisa curve: nonlinear regression (polynomial regression method)

  1. “The box and whiskers plot must be combined with all data points included for clarity. See example attached. Were outliers removed before statistical analysis?”

We thank this recommendation. Plots are done as suggested. Considering the heterogeneity of the disorder, all values were carefully analysed and included in the statistical analysis. The original figures in the supplementary material corroborate the information described above.

  1. Please define the box and whiskers plot in at lease first figure legend.

We thank this recommendation. The following information has been added to the legend of Figures 1 and 2: “…, and their respective scatter plots and bar graph (mean±SD).”

Round 2

Author Response

“Essentially, these are not their role in T cells. This is what happens in the brain/neuronal cell types. Figure 5 is a speculation in T cells. If, yet, there are no described roles for them in T cells in the literature, authors can acknowledge that and this will strengthen manuscript to say it’s the first report to describe this. Line 57 - Peripheral blood nuclear cells (PBMC) also synthetize NE [3]. Please provide reference to original research article, this review does not contain any information about “PBMC synthetize NE”.

We thank this referee for his/her comment. No. In the Figure 5 is clarified some of the common signalling pathways that occurs in all cells in our body (it is biology and not speculation). The three steps of cell signalling are: reception (signaling ligand binds to a receptor), transduction and cellular response (e.g., gene expression). Depending on the ligand, certain signaling cascades will be activated. With figure 5 we try to explain what the absence of complexes implies in some of these signalling cascades that involves NE/NET and NT-3/TrkC. The presence of NET and TrkC on the membrane of T cells has already been described in the literature. What has not been described is the implication of the NET-ext - TrkC immunocomplexes.

The reference in line 3 is incorrect, as are all references in this new paragraph. When refreshing the most recent references in Mendeley, they were not added to the list of references. We apologize for the error. New references are currently listed.

  1. “The specificity of antibody staining must be shown in T cells.”

We thank this referee for his/her comment. We think it refers to the specificity of anti-CD3. Thus, antiCD3 antibody is the T cell specific marker “1:100 anti-CD3 mouse monoclonal antibody FITC conjugated (MA1-10178, Thermo Fisher Scientific, Inc., Waltham, MA, USA)”. The immunostaining of T cells (green) is in the Supplementary Figure 3.

New comment:

“A complete overlap of staining against (1) CD3 and NET and (2) CD3 and TrkC is observed. NET and TrkC are neuronal components, staining images shows NET and TrkC protein expression is completely co-localized and as abundant as CD3, CD3 being highly expressed in T cells, B cells and mononuclear phagocytes. High resolution images of single cells and qCPR expression data to compare CD3 expression with NET and TrkC is desirable and will be extremely elucidative here. Is staining against NT-3 represents secreted NT-3 by PBMCs?”

We thank this referee for his/her comment. CD3 is a marker of T cells (not a B-cell or mononuclear phagocyte marker). A simple analysis of the antibody datasheet specifies such information: “The CD3 subunit complex which is crucial in transducing antigen-recognition signals into the cytoplasm of T cells and in regulating the cell surface expression of the TCR complex.” In rare cases (e.g., B-lineage non-Hodgkin lymphomas), aberrant CD3 expression has been described [1,2]. The marker of B cells is anti-CD19 and immature B cells express CD19, CD20, Cd34 and CD45R. Regarding mononuclear phagocytes, a monocyte marker is, for example, CD14. Attached is a figure from Cell Signaling Technology that clarifies this issue. Thus, there is a co-localization of CD3-NET and CD3-TrkC because CD3 is a membrane protein, NET is a protein with 12 membrane domains, and TrkC is a single transmembrane catalytic receptor. Expression of both NET and TrkC have been described in literature in different non-neural tissues, such as bone marrow, lung, heart, etc. [3–5]. Attached is a figure where Trk/dapi staining was performed in CD34+ bone marrow cells and where the high-expression of receptors in the cell’s membrane is observable.

The aim of this study is to elucidate a mechanism underlying schizophrenia based on the formation of molecular complexes, where CD3 staining was used to confirm the expression of TrkC and NET-ext in the membrane of these cells and therefore images were added to the supplemental material. Regarding the qCPR technique that we assume refers to qPCR, we perform WB to compare levels of proteins expression that are highly stable compared to mRNA.  To clarify the question of using WB and why qPCR would not add significance to the results I quote Dr. Paul A McGettigan (PhD Bioinformatics): “The half-life of different proteins can vary from minutes to days - whereas the degradation rate of mRNA would fall within a much tighter range (2-7hrs for mammalian mRNAs vs 48hrs for protein). The biochemical diversity of proteins means that the individual correlation levels with the associated mRNA are going to vary a lot. ... The transcription data is useful for identifying potential candidates for follow-up work at the protein level. However, in my own experience changes in gene expression level are frequently not reflected at the protein level.”

On the other and, the subcellular localization of NT-3 is extracellular (or secreted). NT-3 is released by different cells (e.g., eosinophiles, lymphocytes, macrophages, neuronal cells, etc.). In this sense, the NT-3 immunostaining marks the NT-3 protein and not its origin.

  1. Pan, Z.; Chen, M.; Zhang, Q.; Wang, E.; Yin, L.; Xu, Y.; Huang, Q.; Yuan, Y.; Zhang, X.; Zheng, G.; et al. CD3-positive plasmablastic B-cell neoplasms: A diagnostic pitfall. Mod. Pathol. 2018, 31, 718–731, doi:10.1038/modpathol.2017.177.
  2. Wang, J.; Chen, C.; Lau, S.; Raghavan, R.I.; Rowsell, E.H.; Said, J.; Weiss, L.M.; Huang, Q. CD3-positive large B-cell lymphoma. Am. J. Surg. Pathol. 2009, 33, 505–512, doi:10.1097/PAS.0b013e318185d231.
  3. Ricci, A.; Felici, L.; Mariotta, S.; Mannino, F.; Schmid, G.; Terzano, C.; Cardillo, G.; Amenta, F.; Bronzetti, E. Neurotrophin and Neurotrophin Receptor Protein Expression in the Human Lung. Am. J. Respir. Cell Mol. Biol. 2004, 30, 12–19, doi:10.1165/rcmb.2002-0110OC.
  4. Cassiman, D.; Denef, C.; Desmet, V.J.; Roskams, T. Human and rat hepatic stellate cells express neurotrophins and neurotrophin receptors. Hepatology 2001, 33, 148–158, doi:10.1053/jhep.2001.20793.
  5. Wehrwein, E.A.; Parker, L.M.; Wright, A.A.; Spitsbergen, J.M.; Novotny, M.; Babankova, D.; Swain, G.M.; Habecker, B.A.; Kreulen, D.L. Cardiac norepinephrine transporter protein expression is inversely correlated to chamber norepinephrine content. Am. J. Physiol. - Regul. Integr. Comp. Physiol. 2008, 295, 857–863, doi:10.1152/ajpregu.00190.2008.

We are grateful for all the reviewer's comments that allowed for substantial improvement of the manuscript.
